# CircDCLRE1C Regulated Lipopolysaccharide-Induced Inflammatory Response and Apoptosis by Regulating miR-214b-3p/STAT3 Pathway in Macrophages

**DOI:** 10.3390/ijms23126822

**Published:** 2022-06-19

**Authors:** Yibin Xu, Yulin Huang, Siyu Zhang, Lijin Guo, Ruiquan Wu, Xiang Fang, Xiaolan Chen, Haiping Xu, Qinghua Nie

**Affiliations:** 1Department of Animal Genetics, Breeding and Reproduction, College of Animal Science, South China Agricultural University, Guangzhou 510642, China; kyrie_xyb@163.com (Y.X.); 15802020639@163.com (Y.H.); zhangsiyu@stu.scau.edu.cn (S.Z.); guolijin2016@163.com (L.G.); wruiquan@163.com (R.W.); direeeeeection@163.com (X.F.); 2Guangdong Provincial Key Lab of Agro-Animal Genomics and Molecular Breeding and Key Lab of Chicken Genetics, Breeding and Reproduction, Ministry of Agriculture, Guangzhou 510642, China; 3State Key Laboratory for Conservation and Utilization of Subtropical Agro-Bioresources & Lingnan Guangdong Laboratory of Agriculture, College of Animal Science, South China Agricultural University, Guangzhou 510642, China; 4School of Life Sciences, Chongqing University, Chongqing 401331, China; xiaolanchend@163.com

**Keywords:** circular RNA, inflammation, apoptosis, miR-214b-3p, STAT3, macrophage

## Abstract

The immune cell inflammation response is closely related to the occurrence of disease, and much evidence has shown that circular RNAs (circRNAs) play vital roles in the occurrence of disease. However, the biological function and regulatory mechanisms of circRNAs in the immune cell inflammation response remain poorly understood. In this study, we constructed an inflammatory model using lipopolysaccharide (LPS)-stimulated chicken macrophage lines (also known as HD11) to verify the function and mechanism of the novel circDCLRE1C (ID: gga_circ_0001674), which was significantly upregulated in spleen tissues infected by coccidia and the macrophage cells exposed to LPS. The results showed that circDCLRE1C aggravated LPS-induced inflammation and apoptosis in HD11 cells. Systemically, circDCLRE1C acted as a sponge for miR-214b-3p binding sites thereby regulating the expression of STAT3. The overexpression of miR-214b-3p rescued the pro-inflammatory effect of circDCLRE1C in HD11 cells stimulated with LPS, and rescued the high expression of STAT3. In conclusion, our study showed that circDCLRE1C could aggravate LPS-induced inflammation and apoptosis through competitive adsorption of miR-214b-3p, thereby increasing the expression of STAT3.

## 1. Introduction

Coccidiosis is considered to be one of the most serious infectious diseases caused by *Eimeria* in modern poultry farming, with huge losses to animal health and the industrial economy [1]. Coccidia invade the intestines of poultry and trigger a devastating inflammatory response that activates the host’s innate and adaptive immune system [2]. As the first line of defense against inflammation, the innate immune response is activated in response to conserved antigenic responses [3]. The production of cytokines such as IL-6 and IFN-γ by pattern recognition receptors that recognize conserved pathogen-associated molecular patterns induces robust innate immunity [4]. These cytokines perform important biological functions in the fight against inflammation and coccidiosis [5,6,7]. The cells involved in innate immune responses to inflammation include natural killer (NK) cells, dendritic cells, and macrophages. In particular, macrophages, also known as inflammatory mediator-secreting immune cells, play a crucial role in mediating innate immune response to inflammation and coccidiosis [8].

Circular RNAs (circRNAs) are novel non-coding RNAs that play an important role in the pathogenesis of immune injury. Compared to linear RNAs, circRNAs are com-posed of a covalently closed continuous loop obtained via back-splicing of precursor mRNAs. Therefore, they are more stable and conserved [9]. Studies have claimed that circRNAs exhibit vital biological functions, including translation into polypeptides, transcriptional regulation, etc. [10,11]. Interestingly, circRNAs can function as sponges, competing for endogenous RNAs (ceRNAs) binding miRNAs to regulate targeted gene expression [12]. Yang et al. showed that circRNA could alleviate pneumonia injury by regulating IGF2 gene expression by sponging miR-24-3p [13]. Moreover, it has been shown that circRNAs have great potential for regulating immune cell functions and inflammatory responses. A previous work found that circRNAs could regulate protein kinase-R (PKR) activation in peripheral blood mononuclear cells (PBMCs) and T cells derived from patients with the autoimmune disease systemic lupus erythematosus [14].

The STAT protein family, which is composed of seven members, plays a pivotal role in transducing signals from the cell surface to the nucleus in response to a wide variety of stimuli. It was demonstrated that they were involved in inflammatory response, the cell cycle, apoptosis, and the regulation of stress-induced gene activation [15,16,17]. Among the STAT protein family, STAT3 played a dominant role in the initiation of stress-induced inflammatory responses and the activation of cytokines when the immune system was damaged [18]. Importantly, many of the downstream pathways of STAT3-encoded cytokines have been proven to dictate cellular responses to stress and inflammation [19]. Emerging studies have shown that STAT3 plays a crucial role in the functions of macrophages [20]. Recent findings have also highlighted that STAT3 participates in immune function through activation of the p38/NF-κB inflammatory signaling axis in macrophages [21]. However, the upstream regulators of STAT3 remain enigmatic.

Our previous study uploaded a comprehensive expression profile of circRNAs, miRNAs, and mRNAs, based on a microarray analysis of spleen gene expression in coccidia-infected Sasso broilers in the laboratory (the dataset is available in the Genome Sequence Archive with the accession number CRA006601). We found the expressions of gga_circ_0001647 and STAT3 were significantly up-regulated, and the expression of miR-214b-3p was significantly down-regulated in the expression profiles (*p* < 0.05) [22]. Gga_circ_0001647 was derived from the pre-mRNA of DNA crosslink repair 1C (DCLRE1C); thus, we termed it circDCLRE1C. In this study, coccidia infection activated chickens’ inflammatory response in the spleen, circDCLRE1C and STAT3 were significantly overexpressed in chicken macrophage inflammation. Therefore, this study was designed to elucidate whether circDCLRE1C is involved in macrophages’ inflammation and function via the miR-214b-3p/STAT3 axis.

## 2. Results

### 2.1. The Characteristics and Subcellular Localization of circDCLRE1C

CircDCLRE1C, or gga_circ_001674, is derived from exon 9 and exon 10 of DNA crosslink repair 1c (DCLRE1C) pre-mRNA, which is located on chromosome 1. Firstly, we designed back-spliced junction (BSJ) primers to amplify circDCLRE1C, and the amplified product was sequenced using Sanger sequencing to validate the circularized junction (Figure 1A and Appendix A). Then, we designed divergent primers to amplify circDCLRE1C, and convergent primers to amplify DCLRE1C, using both complementary DNA (cDNA) and genomic DNA (gDNA). CircDCLRE1C was amplified by divergent primers in cDNA, but not in gDNA (Figure 1B). In addition, the stability and circular nature of circDCLRE1C were verified using the RNase R digestion assay. Further study demonstrated that circDCLRE1C was resistant to RNase R digestion, as shown by Northern blotting (Figure 1C). Nuclear and cytoplasmic separations were performed in HD11 cells to examine the subcellular localization of circDCLRE1C, and showed that the subcellular localization of circDCLRE1C was in the nucleus and the cytoplasm (Figure 1D). Then, we detected the expression of inflammation factors and circDCLRE1C in spleen samples from coccidia-infected and healthy chickens, and found an inflammatory response in the coccidia-infected group, with upregulated levels of IL-6, TNF-α, IFN-γ, and circDCLRE1C compared to the non-infected group (Figure 1E,F). Interestingly, the expression of these inflammation factors (IL-6, TNF-α, and IFN-γ) and circDCLRE1C was dramatically higher in chicken macrophages (HD11) co-cultured with coccidia or LPS, as compared to the control group (Figure 1G). The variable expression of circDCLRE1C in chickens infected with coccidia or exposed to LPS led us to consider whether this was related to inflammation. To investigate the function of circDCLRE1C, we constructed an inflammation model by exposing HD11 cells to LPS

### 2.2. CircDCLRE1C Aggravated Inflammation and Apoptosis in HD11

To explore the role of circDCLRE1C in macrophage mediated inflammation, we performed loss-of-function and gain-of-function experiments to investigate the biological roles of circDCLRE1C in regulating inflammatory signaling cascades. The following experiments were conducted in HD11 cells with or without LPS stimulation. We designed and constructed overexpressing plasmids and siRNA, and we transfected them into normal HD11 cells to detect the efficiency of overexpression and knockdown (Appendix A). We verified the reliability of the pCD25-circDCLRE1C plasmid in HD11 cells with RNaseR treatment (Appendix A) and detected the expression of DCLRE1C to verify that si-circDCLRE1C could not target its linear form (Appendix A). The qRT-PCR assay and ELISA showed that overexpression of circDCLRE1C resulted in upregulated expression of IL-6, TNF-α, and IFN-γ in terms of both transcript and protein levels (Figure 2A), whereas knockdown of circDCLRE1C induced significantly downregulated expression of these inflammatory cytokines (IL-6, TNF-α, and IFN-γ) in the HD11 inflammatory response (Figure 2B). Next, we explored the effects of circDCLRE1C overexpression and inhibition on apoptosis in HD11 cells, via flow cytometry. As shown in Figure 2C, the apoptosis rates of HD11 cells with LPS stimulation significantly increased, and were significantly increased and reduced when circDCLRE1C was overexpressed and knocked-down in the HD11 cells, respectively, during inflammation (Figure 2C,D). In addition, we found that si-circDCLRE1C-mediated downregulation of circDCLRE1C reduced STAT3 mRNA expression while, conversely, circDCLRE1C overexpression upregulated the expression of STAT3 in the LPS-induced inflammation, as determined by qRT-PCR assay (Figure 2E). These results clearly indicate that circDCLRE1C overexpression could aggravate inflammation and apoptosis in HD11 cells.

### 2.3. CircDCLRE1C Could Serve as a Sponge for miR-214b-3p Regulating STAT3 mRNA Expression

In order to determine the mechanism of circDCLRE1C-associated regulation of the progression of macrophage inflammation, we used online software for miRNA target prediction (https://bibiserv.cebitec.uni-bielefeld.de/, accessed on 1 December 2021)) based on a previously uploaded database, and found that miR-214b-3p bound to both circDCLRE1C and the 3′UTR of STAT3 mRNA, and the RNA duplex’s combined mini-mum free energy (MFE) was approximately −21.9 kcal/mol and −27.5 kcal/mol, respectively, indicating that the interaction was more likely (Figure 3A,C). To further validate the miR-214b-3p binding sites on the target circDCLRE1C and STAT3 mRNA, the fragments of circDCLRE1C and the 3′UTR region of STAT3 mRNA containing wild-type or mutant miR-214b-3p binding sites were synthesized and inserted into luciferase reporter vectors. The results showed that the relative luciferase activity was significantly decreased when miR-214b-3p mimics were co-transfected with STAT3 WT or circDCLRE1C WT vectors, compared with the miR-214b-3p mimics and their correspondent mutant reporter co-transfected group (Figure 3B,D). Subsequently, we noted that overexpression of circDCLRE1C or inhibitors of miR-214b-3p led to upregulation of STAT3 mRNA expression, whereas the knockdown of circDCLRE1C or upregulation of miR-214b-3p inhibited the mRNA expression of STAT3 (Figure 3E,F). Overexpression of circDCLRE1 resulted in down-regulation of miR-214b-3p mRNA expression, and knockdown of circDCLRE1C down-regulated miR-214b-3p mRNA expression (Figure 3G). In addition, the greater expression of STAT3 mRNA induced by circDCLRE1C overexpression could be rescued by miR-214b-3p mimics in HD11 cells (Figure 3H). We performed RNA-pulldown experiments in HD11 cells to verify whether miR-214b-3p could physically interact with circDCLRE1C and STAT3, and our agarose gel electrophoresis and qRT-PCR results showed that circDCLRE1C and STAT3 were clearly enriched in the bio-miR-214b-3p mimic group compared with the bio-NC group (Figure 3I,J). Overall, miR-214b-3p could bind to the corresponding sites on circDCLRE1C and STAT3 mRNA, and circDCLRE1C could regulate the mRNA expression of STAT3 by acting as a sponge for miR-214b-3p binding sites.

### 2.4. miR-214b-3p Alleviates Inflammation and Apoptosis in HD11

To better understand the role of miR-214b-3p in macrophage inflammation, we detected the expression of miR-214b-3p in the spleen samples of coccidia-infected chickens, and found that it was significantly reduced, and that the trend was consistent with LPS-induced inflammation in HD11 cells (Figure 4A,B). Subsequently, after transfecting miR-214b-3p mimics and inhibitors into the HD11 cells, we found approximately 100-fold overexpression efficiency of miR-214b-3p mimics and approximately 60% knockdown efficiency of miR-214b-3p inhibitors via qRT-PCR assay (Appendix A). We first studied the effects of miR-214b-3p mimics and inhibitors on HD11 cells with LPS stimulation and found that miR-214b-3p mimics significantly enhanced the expression of IL-6, TNF-α, and IFN-γ compared with the mimic NC group in terms of mRNA and protein levels, as measured by qRT-PCR and ELISA (Figure 4C). On the other hand, the inhibitors of miR-214b-3p significantly reduced the expression of inflammatory cytokines (IL-6, TNF-α, and IFN-γ) (Figure 4D). Moreover, miR-214b-3p mimics and inhibitors also reduced and increased the rates of apoptosis in HD11 cells’ inflammation responses, respectively, as measured by flow cytometry. We found that overexpression and knockdown of miR-214b-3p noticeably downregulated and upregulated the expression and maturity of STAT3 mRNA in HD11 cells with LPS stimulation, respectively (Figure 4E,F). Moreover, we found that miR-214b-3p mimics reduced STAT3 mRNA expression while, conversely, the miR-214b-3p inhibitors upregulated the expression of STAT3 in HD11 cells cultured with LPS, as determined by qRT-PCR assay. In short, miR-214b-3p could suppress the mRNA expression of IL-6, TNF-α, and IFN-γ, as well as apoptosis, in the HD11 cells’ inflammation response.

### 2.5. STAT3 Induced Inflammation and Apoptosis in the HD11

Next, we verified the biological effects of STAT3 mRNA on chicken macrophage inflammation. We found that the expression of STAT3 was significantly increased in HD11 cells cultured with LPS, as well as in the chicken spleen samples infected with coccidia (Figure 5A,B). Then, we designed and constructed the overexpression plasmids and siRNA of STAT3 mRNA, and subsequently transfected them into HD11 cells that were stimulated with LPS. The mRNA expression levels after overexpression and knockdown of STAT3 were detected (Appendix A). The results indicated that overexpression of STAT3 mRNA could noticeably promote the expression of IL-6, TNF-α, and IFN-γ in terms of both transcript and protein levels, as well as increasing the rate of apoptosis (Figure 5CE,F), whereas knockdown of STAT3 could reduce the expression of these inflammatory cytokines significantly, along with the level of apoptosis (Figure 5D–F), during LPS-mediated HD11 inflammation responses. Taken together, the above results demonstrate that STAT3 could play a positive role in HD11 cells’ inflammation.

### 2.6. CircDCLRE1C Aggravated LPS-Induced Inflammation and Apoptosis via miR-214b-3p/STAT3 Pathway

To further investigate whether circDCLRE1C exerts pro-inflammatory effects via the miR-214b-3p/STAT3 pathway in inflammation responses, we co-transfected it with the overexpression vector of circDCLRE1C and miR-214b-3p mimics into HD11 cells exposed to LPS. The results showed that the mRNA expression of STAT3 was significantly increased after upregulation of circDCLRE1C in HD11 cells with inflammatory responses, and this result could be reversed by miR-214b-3p mimics (Figure 6A). Moreover, pCD25-circDCLRE1C-mediated overexpression of circDCLRE1C showed a significant upregulation of the expression levels of IL-6, TNF-α, and IFN-γ in HD11 cells stimulated with LPS (Figure 6B), and the same trend in the rate of apoptosis was also observed (Figure 6C,D).

## 3. Discussion

*Eimeria* is parasitic protozoa that cause chickens’ intestinal tissue to become inflamed and bleed easily after invasion, inducing the host’s innate and adaptive immunity [23,24,25]. Macrophages can be activated by a variety of stimuli, such as parasites, lipoproteins, viruses, and other microorganisms, and are known for their key roles in the immune response [26]. Activated macrophages then play a role in immunomodulatory effects and pathogen clearance [27,28]. The HD11 cells were developed by transforming chicken bone marrow cells with a replication-deficient retrovirus, causing them to function as typical macrophages [29]. In this study, HD11 cells were used as the research object, and the inflammation model induced by LPS stimulation was used to simulate the inflammatory response to infection caused by coccidia invasion. We demonstrated that coccidial challenge could trigger an inflammatory response in the spleen, and a consistent effect of upregulated expression was apparent during coccidial challenge and in LPS-induced macrophages. Then, we used an LPS-induced macrophage inflammation model to investigate the pathobiological effects and mechanisms of circDCLRE1C in vitro. Subsequent in vitro gain-of-function and loss-of-function experiments confirmed that circDCLRE1C could promote macrophage inflammation and promote cell apoptosis.

Numerous studies have shown that circRNAs are closely related to the occurrence of biological diseases, stress response, growth, and development [30,31,32]. Due to the stability of circRNAs, they play an important role in predicting diagnostic markers of disease [33]. At present, circRNAs are divided into three types: exonic circular RNAs, circular intronic RNAs, and exonic–intronic circular RNAs [34]. The biological functions of circRNAs mainly include the following functions: (a) serving as a “sponge” for microRNAs, which act as competitors for endogenous mRNAs; (b) regulating protein binding; (c) regulating gene transcription; and (d) encoding polypeptides. In this study, we obtained the candidate circRNA (gga_circ_0001647), miRNA (miR-214b-3p) and mRNA (STAT3) based on microarray analysis of the spleens of coccidia-challenged broiler chickens and identified the novel gga_circ_0001647 formed by back-splicing of DCLRE1C pre-mRNA, naming it circDCLRE1C. We investigated the structure and mechanism of circDCLRE1C, which showed very high stability and integrity, similar to most other circRNAs—even under RNase R treatment. Studies have reported that the DCLRE1C and STAT3 gene are closely related to the immune function [35,36]. The circRNA acts as a potential upstream regulator of STAT3 gene and indicating that circDCLRE1C may regulate the expression of STAT3. Our results suggest that circDCLRE1C is present in the cytoplasm, and the level of nuclear circDCLRE1C is more than cytoplasmic under normal conditions. However, disruption of cellular homeostasis can drive nuclear circRNAs export [37,38], and circDCLRE1C may act as a sponge adsorbent for miRNAs. Interestingly, we screened for a differential miR-214b-3p belonging to the same family as miR-214-3p, and Cao et al. found that miR-214-3p could regulate the NF-κB pathway to aggravate the progression of osteoarthritis [37]. There have been numerous reports of this family being associated with immune function [38,39]. In addition, it was found that miR-214-3p targets the STAT6 gene to regulate colitis [40], and both STAT3 and STAT6 belong to the same STAT family. Importantly, bioinformatics analysis revealed that circDCLRE1C and STAT3 are a target of gga-miR-214b-3p, and targeting was validated by luciferase reporter assay and RNA pulldown in the HD11 cells. The expression of miR-214b-3p in the spleen was significantly down-regulated after coccidiosis challenge, consistent with LPS-induced inflammation in HD11 cells, exhibiting anti-inflammatory and anti-apoptotic effects. Studies have shown that miRNAs constitute only 3% of the genome, but they are estimated to regulate approximately 90% of genes, each miRNA targets hundreds of mRNAs [41]. The target gene STAT3 of miR-214b-3p showed upregulated expression under coccidial challenge and in LPS-induced macrophages, and the pro-inflammatory and apoptosis-promoting effects were consistent with circDCLRE1C. Importantly, LPS activates a series of signaling pathways and multi-molecular interactions in cells to trigger inflammation and apoptosis, main including STAT, MAPK and NF-κB pathways [42]. We speculate that the circDCLRE1C-STAT3/miR-214b-3p axis is a partly main pathway in the inflammatory response. In brief, this study is the first to find circDCLRE1C to be significantly upregulated in chicken coccidiosis, along with dramatically aggravated macrophage inflammation by sponging miR-214b-3p target sites and modulating STAT3 expression, and that circDCLRE1C is an upstream regulator of STAT3.

The protein STAT3 is a member of the STAT family and plays a key role in pro/anti-inflammatory and apoptotic processes. A previous study found that after the administration of LPS in mice, STAT3 was rapidly activated in the lungs (30 min), and triggered lung inflammation [43]. After inflammatory activation, STAT3 activation was associated with the production of the inflammatory molecules IL-6, TNF-α, and IFN-γ [44,45,46]. These inflammatory factors not only played a key role in the inflammatory response, but also played an important role in the occurrence of parasites. Boyle et al. found that IL-6 was involved in immunoglobulin synthesis and could activate B-cell differentiation in malaria [47]. TNF-α was proven to play a key role in increasing parasite phagocytosis, as well as being involved in parasite control [48]. Elevated serum TNF-α levels have been reported to be directly correlated with the severity of *Plasmodium falciparum* malaria [49]. IFN-γ stimulated the release of pro-inflammatory cytokines involved in the control of infection [50]. Elevated levels or injection of IFN-γ could lead to parasite clearance and/or increase host survival [51]. This study confirmed that STAT3 significantly regulates the expression of cytokines (IL-6, TNF-α, and IFN-γ) and cell apoptosis, and is involved in chicken macrophage inflammation. However, the regulatory mechanism of STAT3 on chicken macrophage inflammation and apoptotic processes through these downstream effectors requires further investigation.

## 4. Materials and Methods

### 4.1. Cell Culture and Sample Sources

The HD11 cell lines were obtained from Susan J. Lamont (Department of Animal Science, Iowa State University, Ames, IA, USA) and Prof. Guobin Chang (Key Laboratory of Animal Genetics and Breeding and Molecular Design of Jiangsu Province, Yangzhou University, Yangzhou, China), and were cultured in Roswell Park Memorial Institute (RPMI) Medium 1640 (Gibco, Carlsbad, CA, USA) with 10% fetal bovine serum, 100 U/mL penicillin, and 100 μg/mL streptomycin at 39 °C in a humidified 5% CO_2_ atmosphere. The spleen tissue from healthy and coccidia-infected chickens was previously stored in the laboratory and originated from Sasso broilers [22].

### 4.2. Transfection

Lipopolysaccharide (LPS) (Sigma, St. Louis, MO, USA) was used to induce inflammation in HD11 cells that were exposed to LPS at a dose of 0.1 μg/mL. Transfections were performed with Lipofectamine 3000 reagent (Invitrogen, Carlsbad, CA, USA) according to the manufacturer’s instructions, with HD11 cells grown to 70–80% confluence. Nucleic acids were diluted in OPTI-MEM Medium (Gibco, Grand Island, NY, USA).

### 4.3. Flow Cytometry Assay

The HD11 cell apoptosis was measured using flow cytometry after staining with an Annexin V–FITC/PI Apoptosis Detection Kit (BD Bioscience, San Jose, CA, USA), according to the manufacturer’s instructions. Briefly, HD11 cells were harvested and washed twice with precooled PBS (Invitrogen, Carlsbad, CA, USA). After suspension with binding buffer, the cells were stained with Annexin V–FITC and propidium iodide (PI) solution for 15 min in the dark at room temperature. Cell apoptosis was analyzed using a FACSCalibur flow cytometer (BD Biosciences, NJ, USA).

### 4.4. Construction of RNA Oligonucleotides and Plasmids

All of the small interfering RNAs (siRNAs)—including siRNA negative controls (si NC), si-circDCLRE1C, si-STAT3—and mimics—including miR-214b-3p mimics, mimic negative controls (mimic NC), miR-214b-3p inhibitors, and inhibitor negative controls (inhibitor NC)—were designed and synthesized by RiboBio (Appendix A, Guangzhou, China). The overexpression vectors in this study, including overexpression of STAT3 (pcD3.1-STAT3) and pmirGLO (pGLO) dual-luciferase reporter vectors, such as pGLO-circDCLRE1C-WT, pGLO-circDCLRE1C-MUT, pGLO-STAT3-WT, and pGLO-STAT3-MUT, were purchased from Tsingke (Beijing, China). The overexpression vector of circDCLRE1C (pCD25-circDCLRE1C) was purchased from Geneseed (Guangzhou, China).

### 4.5. Circular Structure Confirmation

To confirm the stability and characteristics of circDCLRE1C, RNase R (Geneseed Biotech, Guangzhou, China) was used for linear RNA digestion in the total RNA pool obtained from the HD11 cells, according to the manufacturer’s instructions. The control group was treated under RNase-R-free conditions. The back-spliced junction (BSJ) of circDCLRE1C was amplified by a divergent primer, as shown in Appendix A (q-circDCLRE1C.) The amplification products of circRNAs were collected for Sanger sequencing by Tsingke (Beijing, China), and the back-spliced region was analyzed using DNASTAR software (https://www.dnastar.com/, accessed on 4 October 2020).

### 4.6. MiRNA Targets Prediction and RNAhybrid Detection

The target genes of miR-214b-3p were predicted using miRDB (http://mirdb.org/, accessed on 1 December 2021) and RNAhybrid (https://bibiserv.cebitec.uni-bielefeld.de/rnahybrid?id=rnahybrid_view_submission, accessed on 1 December 2021), and the combined minimum free energy (MFE) between miR-214b-3p and the circDCLRE1C or STAT3 3 untranslated regions (3′UTR) was calculated, respectively.

### 4.7. RNA Pulldown

The biotin-labeled miR-214b-3p probe and negative control probes were synthesized by Biosense (Guangzhou, China). The pulldown assay was performed using the microRNA pulldown kit (Biosense, Guangzhou, China) according to manufacturer’s instructions.

### 4.8. Reverse-Transcription Polymerase Chain Reaction (RT-PCR) and Quantitative PCRQ-PCR)

Total cellular RNA was extracted using the RNAiso Plus Kit (Takara, Japan), as directed by the manufacturer. A NanoDrop ND-2000 spectrophotometer (NanoDrop, DE, USA) was used to estimate the total RNA quality and quantity. RT-PCR was performed using an Evo M-MLV RT Premix for qPCR (AG11706, Accurate, Changsha, China) and the Mir-XTM miRNA First-Strand Synthesis Kit (Takara, Dalian, China). Q-PCR was conducted with ChamQ SYBR qPCR Master Mix (Vazyme, Nanjing, China), according to the manufacturer’s protocols. The expression of miR-214b-3p was normalized to that of U6, and the others were normalized to β-actin cDNA. The results of gene expression were determined by the 2−ΔΔCT method. The RT-PCR primers used are shown in Appendix A.

### 4.9. Enzyme-Linked Immunosorbent assay (ELISA)

According to the manufacturer’s instructions for the chicken IL-6, TNF-α, and IFN-γ ELISA Kits (Meimian, Jiangsu, China), the protein concentrations of the interleukins IL-6, TNF-α, and IFN-γ in the culture supernatant of the HD11 cells were determined. The absorbance values (OD) were detected at a wavelength of 450 nm by using a microplate reader (Bio-Tek Instruments, Winooski, VT, USA).

### 4.10. Dual-Luciferase Reporter Assay

This assay was performed using the Dual-Luciferase Reporter Assay Kit (Promega, Madison, WI, USA). Briefly, co-transfection the wild-type (WT) or mutant-type (MUT) pmirGLO vectors revealed circDCLRE1C and STAT3 3′UTR with miR-214b-3p mimics or mimic NC, respectively, for 48 h, as detected using a multifunctional microplate reader (BioTek, Winooski VT, USA). The results are presented as the relative firefly luciferase activity, which was normalized to the activity of renilla luciferase.

### 4.11. Statistical Analysis

All data were expressed as the mean ± standard deviation (SD), with at least three independent replicates. Comparisons between two groups or multiple groups were performed using Student’s *t*-test and one-way analysis of variance, respectively. All of the analyses were performed using GraphPad Prism version 9.0 software (GraphPad Software, La Jolla, CA, USA). Differences were considered significant at * *p* < 0.05; ** *p* < 0.01; and *** *p* < 0.001.

## 5. Conclusions

In summary, we discovered for the first time that circDCLRE1C was significantly upregulated in chicken coccidiosis, and dramatically aggravated macrophage inflammation, leading to increased apoptotic cell death. These data suggest that circDCLRE1C could aggravate LPS-induced inflammation and apoptosis through competitive adsorption of miR-214b-3p, thereby increasing STAT3 expression.

## Figures and Tables

**Figure 1 ijms-23-06822-f001:**
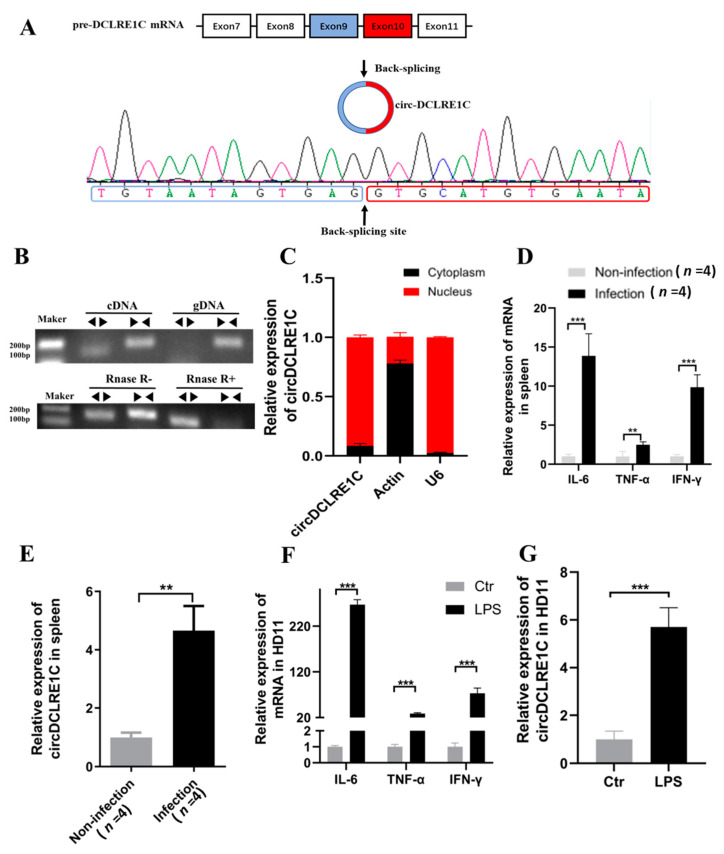
Characteristics of circDCLRE1C: (**A**) The schematic representation of circDCLRE1C’s formation. The back-spliced junction (BSJ) of circDCLRE1C was validated by PCR using a divergent primer, followed by Sanger sequencing. (**B**,**C**) Northern blot analysis for the detection of circDCLRE1C expression in HD11 cells treated with or without RNase R, respectively; divergent primers amplified circDCLRE1C from complementary DNA (cDNA), but not from genomic DNA (gDNA). (**D**) The relative expression levels of circDCLRE1C in the nucleus and cytoplasm were detected by qRT-PCR. (**E**,**F**) The expression of cytokines (IL-6, TNF-α, and IFN-γ) and circDCLRE1C, respectively, in the spleens of non-infected and infected chickens, was detected by qRT-PCR; *n* = 4. (**G**) qRT-PCR analysis of the expression levels of cytokines (IL-6, TNF-α, and IFN-γ) and circDCLRE1C, respectively, induced by LPS stimulation in HD11 cells. Data are presented as means ± SD (*n* = 3 biologically independent samples). ** *p* < 0.01; *** *p* < 0.001 (Student’s *t*-test).

**Figure 2 ijms-23-06822-f002:**
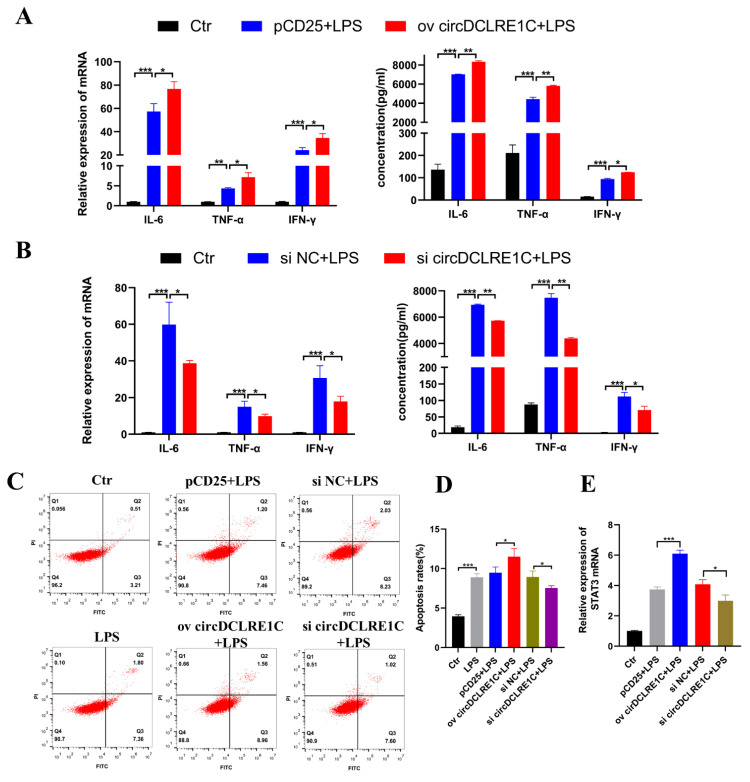
CircDCLRE1C aggravated LPS-induced inflammation and apoptosis in HD11 cells: (**A**,**B**) ELISA and qRT-PCR assay showing the expression levels of IL-6, TNF-α, and IFN-γ in HD11 cells with circDCLRE1C knockdown or overexpression. (**C**,**D**) Cell apoptosis measured with Annexin V–FITC/PI apoptosis detection kit via flow cytometry assay. (**E**) The mRNA levels of STAT3 in HD11 cells with circDCLRE1C overexpression and knockdown. Data are presented as means ± SD (*n* = 3 biologically independent samples). * *p* < 0.05; ** *p* < 0.01; *** *p* < 0.001 (Student’s *t*-test).

**Figure 3 ijms-23-06822-f003:**
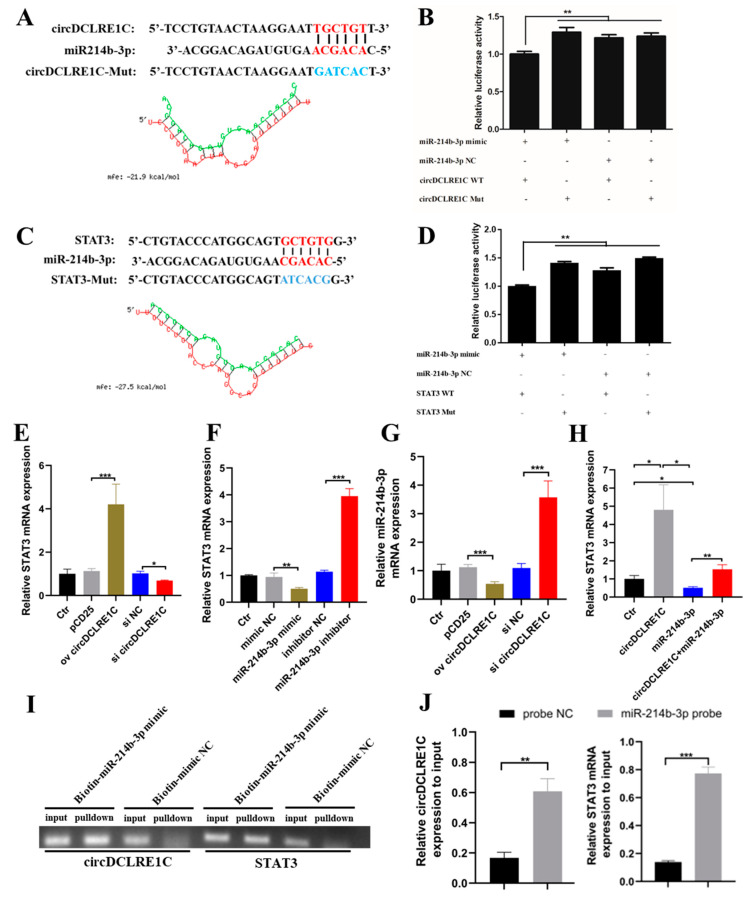
CircDCLRE1C could serve as a sponge for miR-214b-3p to regulate the expression of STAT3: (**A**,**C**) The potential binding site sequence (highlighted in red) of miR-214b-3p on circDCLRE1C or STAT3, and the model of interaction between miR-214b-3p and circDCLRE1C or STAT3, modeled by RNAhybrid software. (**B**,**D**) Dual-luciferase assay verifying the binding relationship between miR-214b-3p and circDCLRE1C or STAT3, respectively. (**E**) The mRNA level of STAT3 in HD11 cells with circDCLRE1C overexpression and knockdown, as measured by qRT-PCR. (**F**) The mRNA level of STAT3 in HD11 cells with miR-214b-3p mimics and inhibitors. (**G**) The mRNA level of miR-214b-3p in HD11 cells with circDCLRE1C overexpression and knockdown. (**H**) The increased expression of STAT3 in HD11 cells with circDCLRE1C overexpression could be rescued by mi-214b-3p. (**I**,**J**) CircDCLRE1C was pulled down by the biotin-miR-214b-3p probe, as detected by gel electrophoresis and qRT-PCR. The relative levels were normalized to the input. Data are presented as means ± SD (*n* = 3 biologically independent samples). * *p* < 0.05; ** *p* < 0.01; *** *p* < 0.001 (Student’s *t*-test).

**Figure 4 ijms-23-06822-f004:**
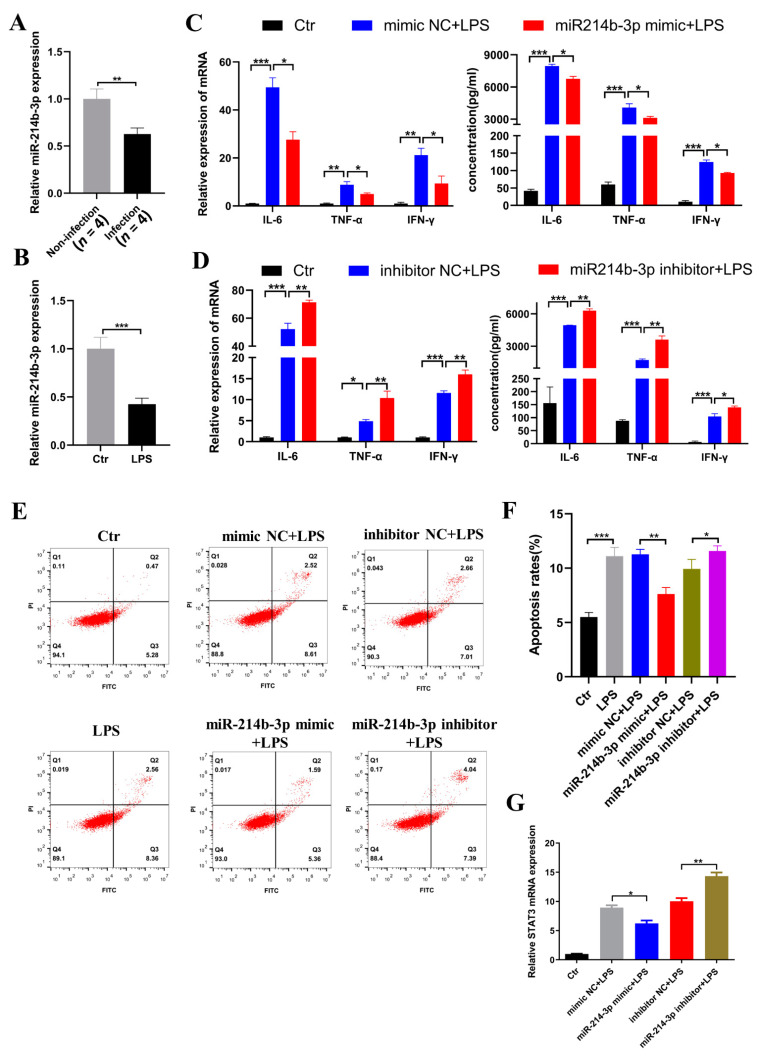
miR-214b-3p alleviated LPS-induced inflammation and apoptosis in HD11 cells: (**A**,**B**) The expression levels of miR-214b-3p were reduced in coccidia-infected samples and the LPS-induced HD11 inflammatory model, as detected by qRT-PCR. (**C**,**D**) The expression levels of IL-6, TNF-α, and IFN-γ in LPS-stimulated HD11 cells with miR-214b-3p mimics and inhibitors, as detected by qRT-PCR and ELISA. (**E**,**F**) Cell apoptosis was detected with Annexin V–FITC/PI double-staining via flow cytometry assay. (**G**) The mRNA levels of STAT3 in HD11 cells with miR-214b-3p mimics and inhibitors. Data are presented as means ± SD (*n* = 3 biologically independent samples). * *p* < 0.05; ** *p* < 0.01; *** *p* < 0.001 (Student’s *t*-test).

**Figure 5 ijms-23-06822-f005:**
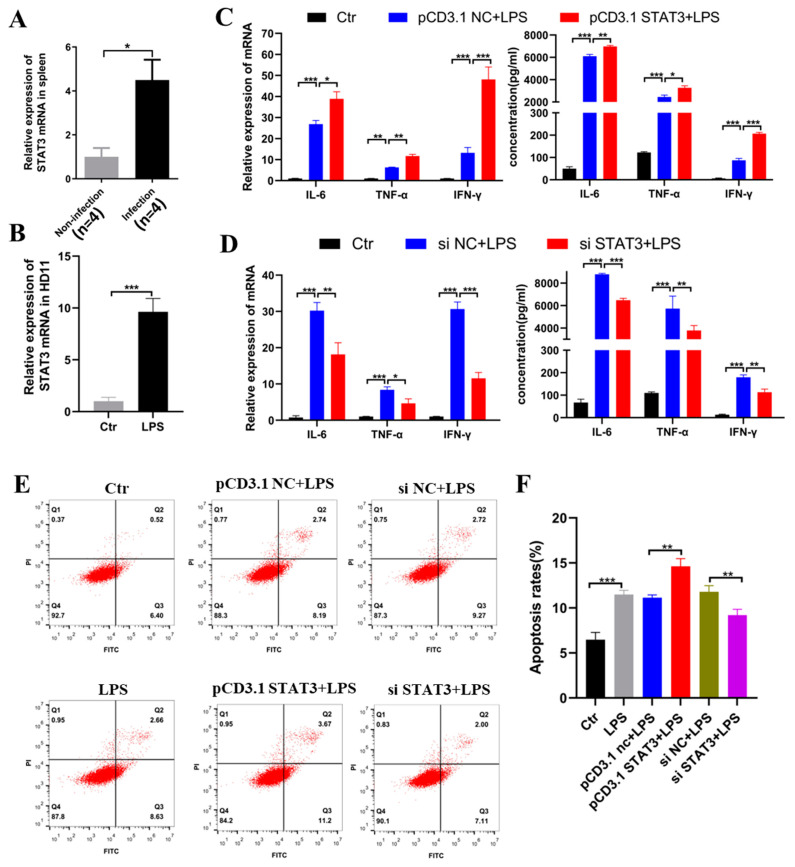
STAT3 aggravated LPS-induced inflammation and apoptosis in HD11 cells: (**A**,**B**) The mRNA expression level of STAT3 was increased in coccidia-infected samples and LPS-induced inflammation in HD11 cells, as detected by qRT-PCR. (**C**,**D**) The expression levels of IL-6, TNF-α, and IFN-γ in LPS-stimulated HD11 cells with STAT3 overexpression and knockdown were detected by qRT-PCR and ELISA. (**E**,**F**) Cell apoptosis was detected with Annexin V–FITC/PI double-staining via flow cytometry assay. Data are presented as means ± SD (*n* = 3 biologically independent samples). * *p* < 0.05; ** *p* < 0.01; *** *p* < 0.001 (Student’s *t*-test).

**Figure 6 ijms-23-06822-f006:**
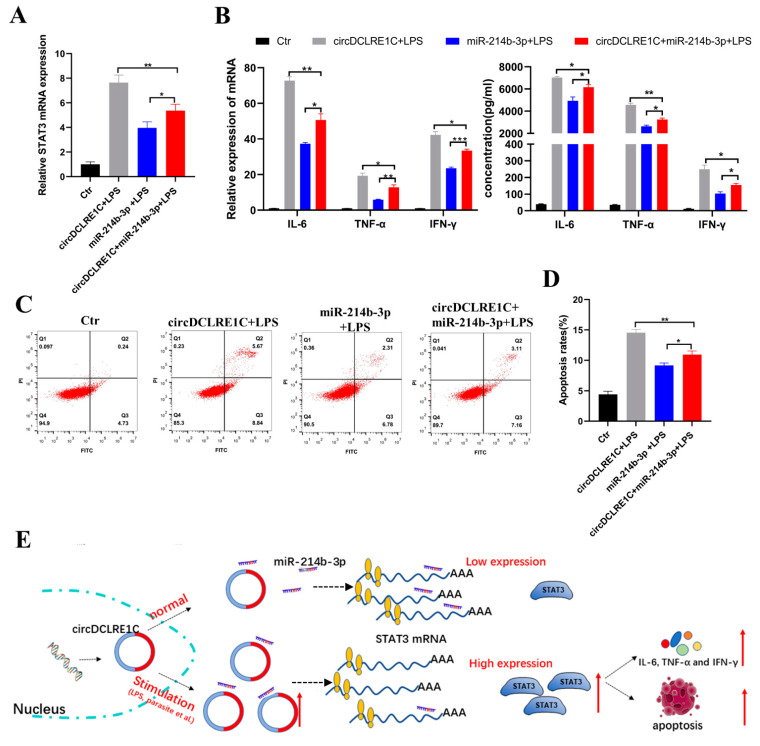
MiR-214b-3p mimics partly reversed the effects of circDCLRE1C in the inflammation responses: (**A**) MiR-214b-3p mimics could rescue the increase in the STAT3 mRNA expression caused by overexpression of circDCLRE1C in HD11 cells stimulated with LPS. (**B**) High expression levels of IL-6, TNF-α, and IFN-γ induced by overexpression of circDCLRE1C were rescued by co-transfection with circDCLRE1C and miR-214b-3p mimics in the inflammatory response. (**C**,**D**) The Ov-circDCLRE1C-induced increase in cell apoptosis was reversed by co-transfection with ov-circDCLRE1C and miR-214b-3p mimics. Data are presented as means ± SD (*n* = 3 biologically independent samples). * *p* < 0.05; ** *p* < 0.01; *** *p* < 0.001 (Student’s *t*-test). (**E**) Graphical abstract: graphical diagram of circDCLRE1C promoting LPS-induced inflammation and apoptosis by sponging miR-214b-3p and regulating STAT3 expression.

## Data Availability

The data presented in this study are available on request from the corresponding author.

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
