# Peer review of "CircDCLRE1C Regulated Lipopolysaccharide-Induced Inflammatory Response and Apoptosis by Regulating miR-214b-3p/STAT3 Pathway in Macrophages"

_ijms, 2022, doi:10.3390/ijms23126822_

Round 1

Reviewer 1 Report

The present manuscript reported the roles of circDCLRE1C and the relationship between circDCLRE1C and miR-214b-3p/STAT3 in the lipopolysaccharide (LPS)-induced macrophage inflammation model. This study arouses interest for readers and provides an important clue to understand circRNAs involved in the mechanisms by which various infection and inflammation diseases occur and progress. However, there are some issues that should be addressed or modified.

(1) According to a previously reported or registered expression profile, authors mentioned that circDCLRE1C and STAT3 were upregulated in coccidian-infected Sasso broilers. Perhaps, numerous mRNAs and miRNAs other than the two molecules would be upregulated. Why did the authors select circDCLRE1C and STAT3 as the key molecules? When authors would like to focus on circDCLRE1C, how did authors select or predict STAT3 as a counterpart of circDCLRE1C? Would you please specify this critical issue in the revised version?

(2) As described by authors, the biological functions of circRNAs include four aspects including competition with miRNAs for endogenous mRNAs (as a “sponge”). Why did authors consider circDCLRE1C as a sponge for STAT3 mRNA? Does circDCLRE1C have other biological functions in the inflammation process? Could you please address the issue in the revised version?

(3) Authors described the reason why HD11 cells were selected in this study. Have authors experienced other macrophage-derived cell lines?

(4) Levels of target molecules by mimics or siRNA/inhibitors were drastically increased or decreased, as shown in Supplementary Figures. However, in comparison to these transitions, changes in the expression levels of IL-6/THF-alpha/IFN-gamma and STAT3 mRNA and apoptosis rates appear to be small. Do these results suggest that the circDCLRE1C–STAT3/miR-214b-3p axis might not be a main pathway in such inflammation? Would you please explain the discrepancy in the revised version?

(5) If circDCLRE1C acts as a sponge for miR-214b-3p, are the miR-214b-3p levels decreased along with an increase in the circDCLRE1C levels?

(6) When STAT3 mRNA is overexpressed or downregulated, how do circDCLRE1C and miR-214b-3p change? Can circDCLRE1C be interpreted as being located upstream of STAT3? Is circDCLRE1C upregulated before STAT3 when inflammation occurs? If so, what is the inflammatory signal for circDCLRE1C?

(7) Would you please specify what are “pCD25” and “pCD3.1”?

(8) Line 65: “Among them, STATs…”; What does the term “them” mean?

(9) Line 229230: This sentence is strange as a summary of this paragraph.

Author Response

Response to Reviewer 1 Comments

Point 1: According to a previously reported or registered expression profile, authors mentioned that circDCLRE1C and STAT3 were upregulated in coccidian-infected Sasso broilers. Perhaps, numerous mRNAs and miRNAs other than the two molecules would be upregulated. Why did the authors select circDCLRE1C and STAT3 as the key molecules? When authors would like to focus on circDCLRE1C, how did authors select or predict STAT3 as a counterpart of circDCLRE1C? Would you please specify this critical issue in the revised version?

Response 1: Thanks for your comments. We obtained that the candidate circDCLRE1C (circ_0001674) and STAT3 from the differential expression profiles of circRNAs, miRNAs, and mRNAs [1] (see figure 1 below). STAT3 plays a major role in the immune response, especially in the pathogenic process and inflammatory response [2], We found that the expression of circDCLRE1C was consistent with STAT3. The parental gene DCLRE1C of circDCLRE1C has also been reported to be associated with immune function [3], There may be a potential relationship between circDCLRE1C and STAT3. The circRNA acts as a potential upstream regulator of STAT3 gene, and indicating that circDCLRE1C may regulate the expression of STAT3. We discuss this issue in the revised version (see line 293-303). In addition, we also studied the roles of other candidate circRNAs and mRNAs, such as circNFIC and DENND1B [4].

Figure 1: Please see the attachment

[1] Chen, X. L.; Wang, Z. J.; Chen, Y. F.; Akinci, I.; Luo, W.; Xu, Y. B.; Jebessa, E.; Blake, D.; Sparks, N.; Hanotte, O. H.; Nie, Q. H., Whole transcriptome sequencing analysis of circRNAs, miRNAs, and mRNAs in Sasso chickens during the challenge of coccidiosis. Front Immunol. 2022. Under review.

[2] Hillmer EJ, Zhang H, Li HS, Watowich SS. STAT3 signaling in immunity. Cytokine Growth Factor Rev. 2016; 31:1-15.

[3] Humblet-Baron S, Schönefeldt S, Garcia-Perez JE, Baron F, Pasciuto E, Liston A. Cytotoxic T-lymphocyte-associated protein 4-Ig effectively controls immune activation and inflammatory disease in a novel murine model of leaky severe combined immunodeficiency. J Allergy Clin Immunol. 2017;140(5):1394-1403.e8

[4] Chen Y, Wang Z, Chen X, Peng X, Nie Q. CircNFIC Balances Inflammation and Apoptosis by Sponging miR-30e-3p and Regulating DENND1B Expression. Genes (Basel). 2021 Nov 19;12(11):1829.

Point 2: As described by authors, the biological functions of circRNAs include four aspects including competition with miRNAs for endogenous mRNAs (as a “sponge”). Why did authors consider circDCLRE1C as a sponge for STAT3 mRNA? Does circDCLRE1C have other biological functions in the inflammation process? Could you please address the issue in the revised version?

Response 2: Thanks for your comments. As the answer to the comments above, we currently focus on the competitive endogenous mechanism, because we predicted that the candidate miR-214b-3p could target circDCLRE1C and STAT3 from the differential expression profiles of circRNA, miRNA, and mRNA. In addition, Cao et al. (2021) reported that miR-214-3p of the same family of miR-214b-3p plays a role in cellular inflammation, and other research reported that miR-214b-3p was associated with the immune functions [6,7]. We speculated that miR-214b-3p might play important roles in cellular inflammation. We addressed the issue in the revised version (see line 303-310).

[5] Cao, Y.; Tang, S.; Nie, X.; Zhou, Z.; Ruan, G.; Han, W.; Zhu, Z.; Ding, C., Decreased miR-214-3p activates NF-κB pathway and aggravates osteoarthritis progression. EBioMedicine 2021, 65, 103283.

[6] Wang, J.; Wang, W. N.; Xu, S. B.; Wu, H.; Dai, B.; Jian, D. D.; Yang, M.; Wu, Y. T.; Feng, Q.; Zhu, J. H.; Zhang, L.; Zhang, L., MicroRNA-214-3p: A link between autophagy and endothelial cell dysfunction in atherosclerosis. Acta Physiol (Oxf) 2018, 222, (3).

[7] Yan, Z.; Zang, B.; Gong, X.; Ren, J.; Wang, R., MiR-214-3p exacerbates kidney damages and inflammation induced by hyperlipidemic pancreatitis complicated with acute renal injury. Life Sci 2020, 241, 117118.

Point 3: Authors described the reason why HD11 cells were selected in this study. Have authors experienced other macrophage-derived cell lines?

Response 2: Thanks for reviewer’s comments. Spleen is one of the central organ in relation to inflammatory response, and it contains a large numbers of immune cells, such as macrophages [8, 9]. HD11 cells have been widely used for the research of infection and immunity in chickens [10, 11]. We could not perform the experiments in other macrophage-derived cell lines, because there was no macrophage-derived cell line in chickens or poultry except for HD11.

[8] A-Gonzalez N, Castrillo A. Origin and specialization of splenic macrophages. Cell Immunol. 2018 Aug; 330:151-158.

[9] Barrea L, Di Somma C, Muscogiuri G, Tarantino G, Tenore GC, Orio F, Colao A, Savastano S. Nutrition, inflammation and liver-spleen axis. Crit Rev Food Sci Nutr. 2018;58(18):3141-3158.

[10] Lin W, Zhou L, Liu M, Zhang D, Yan Y, Chang YF, Zhang X, Xie Q, Luo Q. gga-miR-200b-3p Promotes Macrophage Activation and Differentiation via Targeting Monocyte to Macrophage Differentiation-Associated in HD11 Cells. Front Immunol. 2020,11:563143.

[11] Liniger M, Moulin HR, Sakoda Y, Ruggli N, Summerfield A. Highly pathogenic avian influenza virus H5N1 controls type I IFN induction in chicken macrophage HD-11 cells: a polygenic trait that involves NS1 and the polymerase complex. Virol J. 2012 Jan 9;9:7.

Point 4: Levels of target molecules by mimics or siRNA/inhibitors were drastically increased or decreased, as shown in Supplementary Figures. However, in comparison to these transitions, changes in the expression levels of IL-6/THF-alpha/IFN-gamma and STAT3 mRNA and apoptosis rates appear to be small. Do these results suggest that the circDCLRE1C–STAT3/miR-214b-3p axis might not be a main pathway in such inflammation? Would you please explain the discrepancy in the revised version?

Response 2: Thanks for reviewer’s comments. Compared with these transitions about levels of target molecules by mimics or siRNA/inhibitors, the changes in the expression levels of IL-6, TNF-α, IFN-γ and STAT3 mRNA and apoptosis rates seem to be small. However, the changes around 10%-30% under the strong stimulation of LPS are not small. Studies have shown that miRNAs constitute only 3% of the genome, but they are estimated to regulate approximately 90% of genes, each miRNA targets hundreds of mRNAs [12]. STAT3 is just one of many target genes of miR-214b-3p. Our results showed that circDCLRE1C, miR-214b-3p and STAT3 have significant effects on inflammation. Moreover, LPS activates a series of signaling pathways and multi-molecular interactions in cells to trigger inflammation and apoptosis, main including STAT, MAPK and NF-κB pathways [13], and we speculate that the circDCLRE1C–STAT3/miR-214b-3p axis is a partly main pathway in the inflammatory response. We discuss this issue in the revised version (see line 311-321).

[12] Marques-Rocha JL, Samblas M, Milagro FI, Bressan J, Martínez JA, Marti A. Noncoding RNAs, cytokines, and inflammation-related diseases. FASEB J. 2015 Sep;29(9):3595-611.

[13] Vergadi E, Vaporidi K, Tsatsanis C. Regulation of Endotoxin Tolerance and Compensatory Anti-inflammatory Response Syndrome by Non-coding RNAs. Front Immunol. 2018 Nov 20; 9:2705.

Point 5: If circDCLRE1C acts as a sponge for miR-214b-3p, are the miR-214b-3p levels decreased along with an increase in the circDCLRE1C levels?

Response 2: Thanks for your comment which is valuable for improving the accuracy of the manuscript. We detected the mRNA levels of miR-214b-3p in circDCLRE1C overexpressed and knocked down in HD11 cells by qRT-PCR, and found that overexpression of circDCLRE1 resulted in down-regulation of miR-214b-3p mRNA expression. Meanwhile, knockdown of circDCLRE1C down-regulated miR-214b-3p mRNA expression (see figure 2 below). We supplemented the data to Figure 3G. More information is added in line 168-170 and 187 of revised manuscript.

Figure2: Please see the attachment

Point 6: When STAT3 mRNA is overexpressed or downregulated, how do circDCLRE1C and miR-214b-3p change? Can circDCLRE1C be interpreted as being located upstream of STAT3? Is circDCLRE1C upregulated before STAT3 when inflammation occurs? If so, what is the inflammatory signal for circDCLRE1C?

Response 2: Thanks for your comments. We detected the expression levels of circDCLRE1C and miR-214b-3p by qRT-PCR after STAT3 overexpression or knockdown, and no significant change was found in the expression levels of circDCLRE1C and miR-214b-3p (see the figure 3 below). CircDCLR1C is not located upstream of STAT3, because DCLRE1C and STAT3 genes are located on chicken chromosomes 1 and 27 each. We detected the expression levels of circDCLRE1C and STAT3 in HD11 exposure to LPS for difference time (0h, 0.5h, 1h, 1.5h), and found that the expressions of circDCLRE1 and STAT3 were significantly increased at 1h (see the figure 4 below). We speculated that the expressions of circDCLRE1C and STAT3 concurrently increased when inflammation occurs. In addition, from Figure 3A in the revised version, we found that overexpression of circDCLR1C induced an increased expression of STAT3, and we predicted that one of the upstream pathways of STAT3 was circDCLRE1C and the inflammatory signal was circDCLRE1C.

Figure 3: Please see the attachment

Figure 4: Please see the attachment

Point 7: Would you please specify what are “pCD25” and “pCD3.1”?

Response 2: Thanks for your comment. pCD25 is the name of a circular RNA overexpression vector, synthesized by a Geneseed biotechnology company (Guangzhou, China). PCD3.1 is the abbreviation of eukaryotic overexpression vector pCDNA3.1. The information of vectors is shown in figure 5 below.

Figure 5:Please see the attachment

Point 8: Line 65: “Among them, STATs…”; What does the term “them” mean?

Response 2: Thanks for your comment. Our writing was wrong. “them” represents STAT protein family. The original “them” represents STAT3 not STATs. We have corrected in revised version. More information see line 66-67.

Point 9: Line 229230: This sentence is strange as a summary of this paragraph.

Response 2: Thanks for reminding, we agree with you, and have corrected the summary of this paragraph. More information is added in line 237-238 of the revised version.

Reviewer 2 Report

Review of IJMS-1710003V1

CircDCLRE1C Aggravated Lipopolysaccharide-Induced Macrophage Inflammation and Apoptosis by Regulating miR-214b-3 3p/STAT3 Expression

This was a difficult manuscript to read. Many of the groupings of words used in the textstream are inappropriate (see my comments below) which leads to the difficulty in reading and comprehension of the text put forward by the authors. The text needs a thorough re-write, the figures also are far far too small- these need improvement.  The standard statistical term is P CAP Italics ie P and not small p

Title

The title appears incomplete and does not really make sense -seems like there are some words missing-PLEASE CORRECT THIS

Abstract

The authors need to state the aim of their study in the abstract and a clear conclusion. The wording in the abstract needs to be tightened up-some comments need re-wording or correct English inserted. Eg many evidences ? in the immune cell inflammation expression of the STAT3.

Line 42 “pattern receptors”-pattern recognition receptors ?

Line 42-47 reword in correct English

Line 50 “immune injury.”-maybe could be reworded better WHAT IS IMMUNE-INJURY?

Line 51 “Therefore, they are more stable and conservation [9].”- not a sentence

Line 59 define PKR

Line 60 “T cells derived from immunity disease systemic lupus erythematosus” re-word

Line 64 “inflammatory responses,”-not plural

Line 68 “of the downstream target genes of STAT3 encoded cytokines and growth factors were proved to dictate cellular responses to stress and inflammation”-re-word more clearly

Line 70 “role in the macrophage functions”-re-word

Line 80 “In this study, chicken activated spleen inflammatory response by coccidia infected, and interestingly, circDCLRE1C and STAT3 were also significantly overexpressed in chicken macrophage inflammation” re-word

Fig 1 legend define BSJ , gDNA I presume this is genomic DNA, “of non-infection or infection chickens were”- NON-INFECTED OR INFECTED CHICKENS, “QRT-PCR analysis the expression levels of cytokines”-RE-WORD

Line 118 “biologically independent samples” ?

Line 120 “CircDCLRE1C aggravateg inflammation and apoptosis in HD11”-correct the wording -why italics?

Line 121 “in the macrophage mediated inflammation”

Line 127 “up-regulated expression”-up-regulation?

Line 129 “down-regulated expression”-down-regulation?

Line 130 “inflammation cytokines”-inflammatory cytokines ?

Line 130 “significantly in the HD11 inflammatory response “   re-word

Line 133 “LPS stimulation were significantly rise” elevated?

Line 133/134 “and significantly increased and reduced when circDCLRE1C was overexpressed and knockdown in HD11 inflammation, respectively” re-word with correct English.

Line 135 “we found that si-circDCLRE1C-mediated down-regulation of circDCLRE1C reduced STAT3 mRNA expression, and while conversely circDCLRE1C overexpression up-regulated the expression of STAT3 in the LPS 137 induced-inflammation by qRT-PCR assay” re-word using correct English.

Figure 2 is very small-some of the numbers are not intelligible. If this figure was presented in portrait rather than landscape format then individual parts of the figure could be presented at a larger size where all components were discernable. Text sizes should be uniform throughout the figure. The figure legend should be more informative.

Line 150 “phage inflammation progression,”-what does this mean

Line 150-165 re-word

Line 169 “were clearly enrich” enriched

Figure 3 is too small-see my earlier comments for Figure 2. The legend needs re-wording

Line 191 “Subsequently, after transfected miR-214b-3p mimic and inhibitor into HD11, we found approximate a hundred folds of over-express efficiency”-re-word to make sense

Figures 4-6 are all far too small-the reader should not be expected to use a magnifying glass to discern all of the information in the figures. A portrait format for the figures would be more appropriate and would allow component parts to be presented at a larger magnification.

Author Response

Thank you for taking time out of your busy schedule to review the manuscript. Now we have carefully corrected and replied the manuscript for this revision. The revision instruction are as follows:

This was a difficult manuscript to read. Many of the groupings of words used in the textstream are inappropriate (see my comments below) which leads to the difficulty in reading and comprehension of the text put forward by the authors. The text needs a thorough re-write, the figures also are far far too small- these need improvement. The standard statistical term is P CAP Italics ie and not small p

Answer: We appreciate very much for your suggestion, and we have done it according to your ideas. We corrected the grammatical errors, spelling and grammar errors as shown below, and polished the whole manuscript. We would like to confirm that the suitably revised manuscript is understandable to readers. In addition, the figures were changed from small to large, and the statistical term p was changed to P

Title

The title appears incomplete and does not really make sense -seems like there are some words missing-PLEASE CORRECT THIS

Answer:  Thanks for reminding. We have corrected to “CircDCLRE1C Regulated Lipopolysaccharide-Induced Macrophage Inflammation and Apoptosis by Regulating miR-214b-3 3p/STAT3 Pathway”.

Abstract

The authors need to state the aim of their study in the abstract and a clear conclusion. The wording in the abstract needs to be tightened up-some comments need re-wording or correct English inserted. Eg many evidences ? in the immune cell inflammation expression of the STAT3.

 Answer: Thanks for your suggestion, we have corrected. (See line 19-32)

Line 42 “pattern receptors”-pattern recognition receptors ?

Answer: Thank you for suggestion, the observation is correct, we have change. (See line 42)

Line 42-47 reword in correct English

Answer:  Thank you for suggestion, we have corrected. (See line 42-48)

Line 50 “immune injury.”-maybe could be reworded better WHAT IS IMMUNE-INJURY?

Answer: Thank you for comment. Tissue damage caused by cellular or humoral immune responses to endogenous or exogenous antigens is called immune injury.

Line 51 “Therefore, they are more stable and conservation [9].”- not a sentence

Answer: Thanks for reviewer's suggestion and we have corrected. (See line 52)

Line 59 define PKR

 Answer: As for the referee’s concern, the full descriptions of the abbreviations has been supplemented in the revised manuscript (See line 59)

Line 60 “T cells derived from immunity disease systemic lupus erythematosus” re-word

 Answer: As for the referee’s concern and we have corrected. (line 60-61)

Line 64 “inflammatory responses,”-not plura

 Answer: Thanks for your suggestion. We have corrected. (line 64)

Line 68 “of the downstream target genes of STAT3 encoded cytokines and growth factors were proved to dictate cellular responses to stress and inflammation”-re-word more clearly

 Answer: Thanks for your suggestion. We have corrected. (line 68-69)

Line 70 “role in the macrophage functions”-re-word

 Answer: Thank you for suggestion, and we have corrected. (See line 70)

Line 80 “In this study, chicken activated spleen inflammatory response by coccidia infected, and interestingly, circDCLRE1C and STAT3 were also significantly overexpressed in chicken macrophage inflammation” re-word

 Answer: Thank you for suggestion, and we have corrected. (See line 81-83)

Fig 1 legend define BSJ , gDNA I presume this is genomic DNA, “of non-infection or infection chickens were”- NON-INFECTED OR INFECTED CHICKENS, “QRT-PCR analysis the expression levels of cytokines”-RE-WORD

 Answer: Thank you for suggestion, and we have corrected. (See line 112-122)

Line 118 “biologically independent samples” ?

 Answer: Thank you for suggestion, biologically independent samples mean three independent replications.

Line 120 “CircDCLRE1C aggravateg inflammation and apoptosis in HD11”-correct the wording -why italics?

 Answer: Thank you for suggestion, and we have corrected italics are required in the template. (See line 123)

Line 121 “in the macrophage mediated inflammation”

 Answer: Thank you for suggestion, and we have corrected. (See line 124)

Line 127 “up-regulated expression”-up-regulation?

 Answer: Thank you for suggestion, and we have corrected. (See line 131)

Line 129 “down-regulated expression”-down-regulation?

 Answer: Thank you for suggestion, we have corrected. (See line 130)

Line 130 “inflammation cytokines”-inflammatory cytokines ?

 Answer: Thank you for suggestion, and we have corrected. (See line 133)

Line 130 “significantly in the HD11 inflammatory response “   re-word

 Answer: Thank you for suggestion, and we have corrected. (See line 134)

Line 133 “LPS stimulation were significantly rise” elevated?

 Answer: Thank you for suggestion, and we have corrected. (See line 137)

Line 133/134 “and significantly increased and reduced when circDCLRE1C was overexpressed and knockdown in HD11 inflammation, respectively” re-word with correct English.

 Answer: Thank you for suggestion, and we have corrected. (See line 137)

Line 135 “we found that si-circDCLRE1C-mediated down-regulation of circDCLRE1C reduced STAT3 mRNA expression, and while conversely circDCLRE1C overexpression up-regulated the expression of STAT3 in the LPS 137 induced-inflammation by qRT-PCR assay” re-word using correct English.

 Answer: Thank you for suggestion, and we have corrected. (See line 139-141)

Figure 2 is very small-some of the numbers are not intelligible. If this figure was presented in portrait rather than landscape format then individual parts of the figure could be presented at a larger size where all components were discernable. Text sizes should be uniform throughout the figure. The figure legend should be more informative.

 Answer: Thank you for suggestion, and we have corrected. (See Figure 2)

Line 150 “phage inflammation progression,”-what does this mean

 Answer: Thank you for suggestion, it means the progression of macrophage inflammation and we have corrected. (See line 153)

Line 150-165 re-word

 Answer: Thank you for suggestion, and we have corrected. (See line 153-165)

Line 169 “were clearly enrich” enriched

 Answer: Thank you for suggestion, and we have corrected. (See line 175)

Figure 3 is too small-see my earlier comments for Figure 2. The legend needs re-wording

 Answer: Thank you for suggestion, and we have corrected. (See Figure 3)

Line 191 “Subsequently, after transfected miR-214b-3p mimic and inhibitor into HD11, we found approximate a hundred folds of over-express efficiency”-re-word to make sense

 Answer: Thank you for suggestion, and we have corrected. (See line 197-198)

Figures 4-6 are all far too small-the reader should not be expected to use a magnifying glass to discern all of the information in the figures. A portrait format for the figures would be more appropriate and would allow component parts to be presented at a larger magnification.

Answer: Thank you for suggestion, and we have corrected. (See Figure 4-6)

Round 2

Reviewer 2 Report

The authors have not adequately responded to my earlier critiques.

The title remains inadequate and does not make sense.

Author Response

Dear Reviewer:

Thank you for taking time out of your busy schedule to review the manuscript. The revision instruction are as follows:

Point 1: The authors have not adequately responded to my earlier critiques.The title remains inadequate and does not make sense.

Response 1: Thank you for suggestion. We did a point-by-point response of your review before. We have carefully considered the title and corrected to “CircDCLRE1C Regulated Lipopolysaccharide-Induced Inflammatory Response and Apoptosis by Regulating miR-214b-3p/STAT3 Pathway in Macrophages”